# Systematic Review on New Challenges of University Education Today: Innovation in the Educational Response and Teaching Perspective on Students with Disabilities

María Dolores Pérez-Esteban *[ID], Jose Juan Carrión-Martínez and Luis Ortiz Jiménez [ID]

Education Department, University of Almería, 04120 Almeria, Spain; jcarrion@ual.es (J.J.C.-M.); lortizj@ual.es (L.O.J.)

* Correspondence: mpe242@ual.es

**Abstract:** Offering a suitable educational response to students with disabilities continues to be a challenge for higher education institutions, where the teaching attitude, the educational strategies to deal with diversity and the obstacles or difficulties continue to condition the commitment to inclusive education at universities. This systematic review has been carried out following the indications of the PRISMA method. The main objective is to present critical information about the educational response offered to students with disabilities at higher education institutions. Fourteen articles dealing with teaching attitude, difficulties and strategies used were reviewed. The results show how there is a positive attitude towards students with disabilities in some areas, various inclusive strategies are established in the classroom and there are still obstacles that make it difficult to meet all students' needs. In conclusion, inclusive education continues to be a pending issue for university institutions, which are moving towards inclusion, yet at a slower pace, in comparison to other education levels.

**Keywords:** disability; university; teaching–learning process; methodology; social challenges

## 1. Introduction

The main objective of higher education is to deepen academic knowledge, strengthen scientific research and meet the needs of all students (Riddell and Weedon 2014). Establishing inclusive education in all educational institutions continues to be an objective present in the educational panorama (Ainscow 2020), both at the national (Sarrionandia 2017) and international levels (Aiello et al. 2019).

This commitment is supported by international institutions throughout the multitude of declarations, resolutions and actions that the United Nations has launched to achieve this goal. Thus, from the World Declaration on Education for All (UNESCO 1990), the Convention on the Rights of Persons with Disabilities (CRPD 2016), the World Conference on Special Educational Needs: Access and Quality (UNESCO 1994) and the World Conference on Higher Education (UNESCO 1998), there have been many efforts to achieve the inclusion of people with disabilities.

This is reflected in the educational policies of countries with extensive experience in developing the principles of equal opportunities, universal accessibility and non-discrimination for people with disabilities, as is the case of Spain (Díaz 2021), the United Kingdom (Bunbury 2018), Argentina (Precci 2021), Albania (Sulaj et al. 2021), Italy (de Anna and Utge 2019) and Portugal (Ferreira et al. 2015). However, it has been observed how these educational advances are increasingly a reality of social change, since in countries such as Poland (Zielińska 2022), Chile (Paz Maldonado 2020) and Australia (Kent et al. 2018), a process of implementation of inclusive education has begun in the university contexts; these countries are pioneers in achieving advances in this area, eliminating barriers to learning and promoting the participation of all people in educational and vocational activities. Although progress continues towards the promotion of a real inclusive education within the university context, there are

still disparities in terms of the implementation of various educational or universal access strategies that promote access and participation of this group (European Commission 2021).

The university reality differs from this legislative ideal, despite all the efforts that have been made in the educational stages at the primary and secondary levels. Novo Corti and Cantero (2012) observe how face-to-face higher education is a hostile environment for people with disabilities since there are still architectural and academic barriers that hinder participation under equal conditions and the development of real inclusion. Despite the fact that the number of students with disabilities has increased considerably (EADSNE 2011) as observed in countries such as Australia (Department of Education and Training 2016), the United Kingdom (Sachs and Schreuer 2011) and the United States (Allen and Seaman 2014), this group of students is still underrepresented in higher education (Lang 2013). However, these data must be analyzed with caution since there may be great dissonances between the statements of the students and the real data from the universities of each country (Rodríguez Martín et al. 2014).

In this situation, it is not surprising that many of these students choose to pursue their studies in distance institutions that have considerably increased the number of students with disabilities, reaching 3 times (Kotera et al. 2019). It has been shown that virtual environments help students with disabilities to overcome some of the main barriers that limit them from accessing face-to-face higher education (Verdinelli and Kutner 2016), such as stigmatization, accessibility problems and the perception of low ability on the part of their peers (Akin and Huang 2019).

Universities cannot and should not remain behind in the educational advances that are taking place in terms of inclusion (Rodríguez Martín et al. 2014). Inclusive pedagogy is a theoretical–practical approach that starts from four main dimensions, namely beliefs, knowledge, design and actions (Gale et al. 2017), and that makes the university accessible to all (Arini 2020).

Although the research on disability that has been carried out in the field of higher education is not as abundant, it is compared with the work carried out in the previous stages. Studies on student access (Bastías et al. 2020), on attitudes and participation in university life (Lightfoot et al. 2018), on accessibility (Kent et al. 2018) and on teacher training (Banks 2019) stand out. All these aspects condition, in one way or another, the educational response offered to people with disabilities.

Educational response is understood as all those actions that, within the principles of the inclusive school, take into account every student and act in accordance with their specific needs, adopting measures and resources that make it possible to access and remain within the educational system with equal opportunities (Banks 2019). The different educational and social changes that are being developed in favor of inclusive education have had an impact on the teaching role (Polo Sánchez et al. 2021), on the strategies used to improve teaching–learning processes (Lledó et al. 2020) and on the elimination of the different barriers and obstacles that prevent students from entering higher education (Sulaj et al. 2021).

Together with the educational response that is offered to students with disabilities and focusing on the teaching role, the attitudes developed by university professors favor or hinder the implementation of inclusive education. The study carried out by Gibson (2014) shows that the negative attitude on the part of teachers was one of the most difficult barriers to overcome. In line with the results of this study, the genesis of the negative attitude lies in the lack of knowledge and lack of training of university teachers to be able to carry out the necessary adjustments in matters of attention to disability (Lightfoot et al. 2018; Banks 2019). This supposes an added difficulty to be able to attend to these students (Lombardi et al. 2013). Polo Sánchez et al. (2021) show that teachers' attitudes will be more positive at a higher level of information about disability during the university training process.

Facilitating the inclusion of students with disabilities in the classroom is closely linked to the development of inclusive strategies in the teaching–learning processes, in which reasonable adjustments can be made whenever necessary. Reasonable adjustment is understood as each of the adaptations and modifications that at a physical, attitudinal or

social level are necessary to treat the specific needs of people with disabilities (Díaz 2021). Various studies (Bunbury 2018; Fichten et al. 2016) show that these reasonable adjustments would not be necessary if the subjects and teaching projects were planned and designed considering the principles of universal learning design (Tobin 2014). The universal design makes it possible to design the teaching–learning processes attending to each one of the students (Rodríguez Martín et al. 2014; Arini 2020; Bastías et al. 2020).

Thus, the barriers which students with disabilities must face are related to factors external to their condition and are focused on learning environments (McManus et al. 2017). Two types are identified. The first type is traditional architectural barriers such as classrooms, stairs, inadequate auditoriums, heavy doors, broken elevators or the absence of ramps and signs, which continue to pose a handicap when it comes to accessing face-to-face higher education (García-González et al. 2021). Difficulties are ratified in the study developed by Danso et al. (2017) which shows that architectural barriers increase the discrimination suffered by this group.

Academic barriers are becoming a new challenge to face. The teaching attitude, the refusal to adopt various educational strategies and the limitations when accessing the contents and materials of the subjects are some of the obstacles that must be overcome to guarantee inclusive quality education in the university environment (Morgan 2021). Difficulties were detected in the study developed by Moriña et al. (2020), in which they stated that many of the curricula are not inclusive and inferred the teaching attitude towards disability.

To carry out this study, we start from an approach based on the fact that all educational systems assume the challenge of providing quality education in which attention to the diversity present in the classroom is established as a key element to promote an inclusive education and be able to extrapolate it to today's society (Azorín Abellán et al. 2017). Attention to students with disabilities is conceived as an essential requirement to be achieved as part of the Sustainable Development Goals promulgated by the United Nations General Assembly for the 2030 Agenda (De la Rosa Ruiz et al. 2019).

Therefore, the following research questions are raised: (1) What is the attitude of university teachers about the need to provide an educational response to students with disabilities? (2) What innovative strategies do university professors use to develop teaching–learning processes with students with disabilities in university classrooms? (3) What are the main obstacles when dealing with students with disabilities in universities? These three questions are linked to the more general objective of this study, which is to present critical information on the educational response offered to students with disabilities in higher education institutions.

This general objective is broken down into the following specific objectives: (1) analyze current attitudes of university teaching staff after years of inclusive processes; (2) identify innovative strategies developed by university professors to carry out the teaching–learning processes with students with disabilities in university classrooms; (3) describe what remain the main obstacles or difficulties in dealing with students with disabilities in universities.

## 2. Materials and Methods

To carry out the following systematic review, the Preferred Reporting Items for Systematic Reviews and Meta-Analyses (PRISMA) guidelines (Page et al. 2021; Liberati et al. 2009) have been followed. The main objective is to select those studies and research related to the educational response offered to students with disabilities in higher education institutions.

The design and execution of this systematic review consist of several phases that are presented below.

### 2.1. Phase 1: Search Strategies

In the first place and for the development of this first phase, a review of the literature has been carried out in a general way in the main scientific research databases and specialized journals in the field of educational sciences, with national and international character,

with the firm purpose of being able to cover the largest number of investigations that are related to the objectives set out above.

The databases used have been selected based on their relationship, suitability, and their relevant and significant nature within the field of educational sciences. These databases are Web Science (WoS), Scopus and Dialnet Plus.

During the execution of this phase and to carry out the search in these databases, the following descriptors were established: in Spanish, "discapacidad", "estudiante con discapacidad", "universidad", "educación superior", "procesos de enseñanza-aprendizaje", "metodologías", "programas", "innovación", "intervención", "obstáculos" and "dificul-tades"; in English, "disability", "student with disability", "university", "higher education", "teaching-learning processes", "methodologies", "programs", "innovation", "intervention", "obstacles" and "difficulties" (Figure 1).

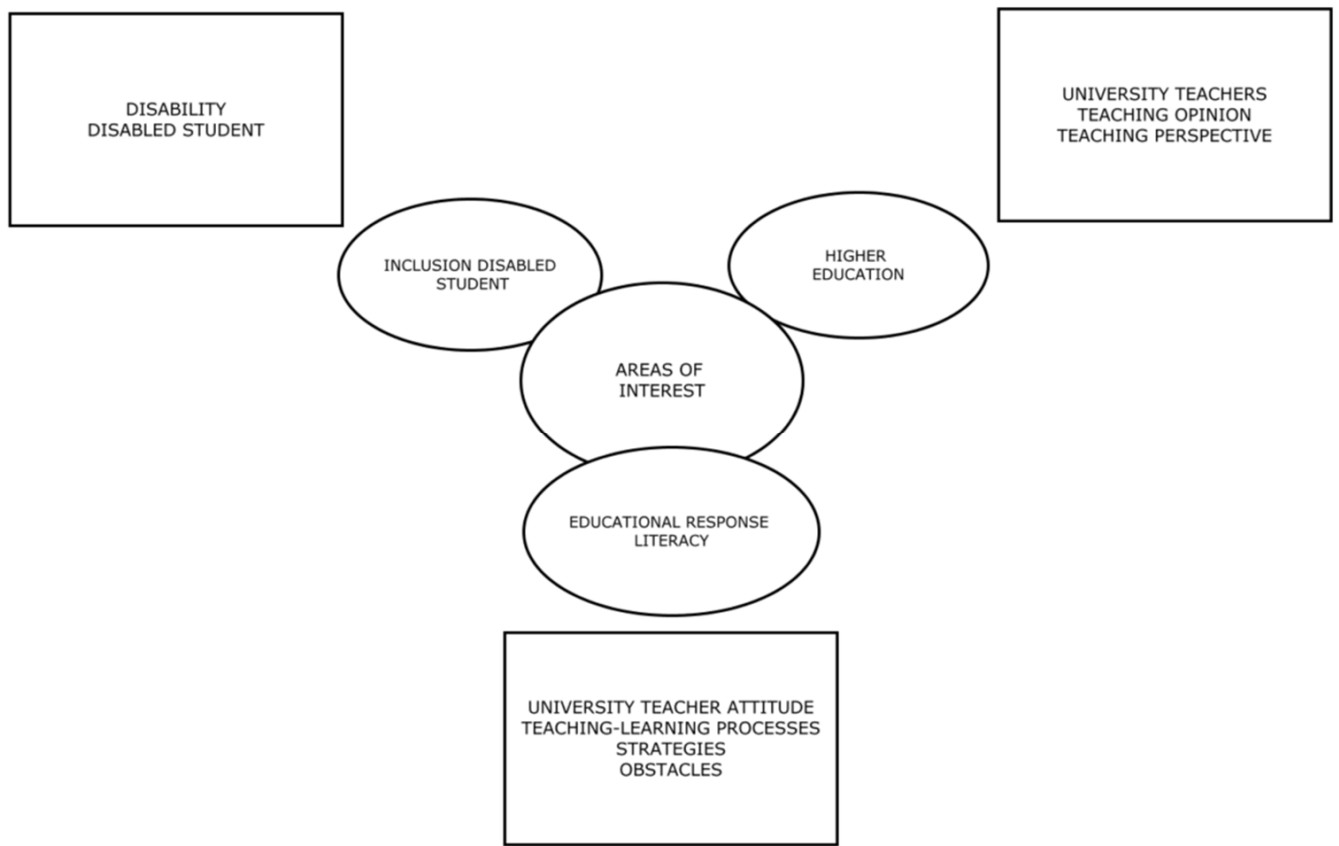

**Figure 1.** Diagram of search terms used in the systematic review.

Both the use of descriptors in both languages (English and Spanish) and the use of the selected databases are supported due to the national and international nature of this research.

Continuing with the development of this phase, secondly, the search was carried out in each of the databases using filters that turned out to be the most appropriate in each case in order to narrow the search to the research topic.

In the third and last place, the review of the titles, abstracts and keywords was carried out, thus making the first selection of the data based on the results obtained after applying the main descriptors "disabled student", "university" and "teaching-learning process" that were used accompanied by some of those mentioned above and using the inclusion and exclusion criteria detailed below.

## 2.2. Phase 2: Inclusion and Exclusion Criteria

The second phase contemplates the inclusion and exclusion criteria that have been used in carrying out this systematic review. Regarding inclusion criteria, investigations chosen were those in which three of the main descriptors or their variants were present in the title, the abstract or the keywords of the selected works, for which the year of publication was between 2018 and 2022, and which were written in Spanish or English, and it was also necessary for the study to have open access to the full text. Regarding exclusion criteria, works that were not related to the educational response offered to students with disabilities at a university, research on this subject in educational stages other than higher education, articles that had no relationship with the investigated topic, papers that did not have access to the full text, non-empirical studies and all papers in which the research is based on the explanation of a future study were excluded.

## 2.3. Phase 3: Screening and Selection Process

In this third phase, the screening and selection process of the works resulting from the search was carried out; this screening, which was carried out from March 2022 to June 2022, was performed by two independent reviewers and supervised by a third reviewer to be able to resolve any disagreement in the selection of studies according to the inclusion and exclusion criteria mentioned above.

The final number of documents used in this review was a total of 14 works, among which a homogeneous finding is observed in the three databases used. Of the total sample, 92.85% (*n* = 13) of the works are written in English, while only the remaining 7.14% (*n* = 1) is written in Spanish.

Next, Figure 2 is presented, where the search scheme of the different investigations can be seen, and Table 1 offers a detailed description of the searches that have been carried out, showing the descriptors with the Boolean operators, the filters applied and the number of articles that have been selected in each of the three stages of this review.

**Table 1.** Procedure for selecting articles from the structured search in the primary databases.

| Database | Boolean Operations | Initial Number | Filters | After Filters | After Criteria | Final |
|---|---|---|---|---|---|---|
| WoS | disabilit * | 178,137 | Domain: social sciences; databases: Web of Science Core Collection; languages: English, Spanish; research areas: education educational research, special education | 2710 | 25 | 5 |
| | disabled student * AND (university OR higher education) | 2196 | | 57 | | |
| | disabled student * AND (teacher attitude) AND (university OR higher education) | 74 | | 6 | | |
| | disabled student * AND teaching-learning process * AND innovation AND (university OR higher education) | 7 | | 0 | | |
| | disabled student * AND (difficult * OR obstacle *) AND (university OR higher education) | 314 | | 13 | | |
| | disabled student * AND (methodolog * OR program * OR intervention) AND (university OR higher education) | 763 | | 22 | | |
| | disabled student * AND (educational response) AND (university OR higher education) | 96 | | 2 | | |

**Table 1.** *Cont.*

| Database | Boolean Operations | Initial Number | Filters | After Filters | After Criteria | Final |
|---|---|---|---|---|---|---|
| SCOPUS | disabilit * | 441,302 | Search within: article title, abstract, keywords; years: 2018–2022; subject area: social sciences (exclude the rest); languages: English, Spanish | 8863 | 31 | 5 |
| | disabled student * AND (university OR higher education) | 1631 | | 108 | | |
| | disabled student * AND (teacher attitude) AND (university OR higher education) | 74 | | 7 | | |
| | disabled student * AND teaching-learning process * AND innovation AND (university OR higher education) | 2 | | 0 | | |
| | disabled student * AND (difficult * OR obstacle *) AND (university OR higher education) | 96 | | 9 | | |
| | disabled student * AND (methodolog * OR program * OR intervention) AND (university OR higher education) | 678 | | 69 | | |
| | disabled student * AND (educational response) AND (university OR higher education) | 38 | | 12 | | |
| Dialnet plus | disabilit * | 12,508 | Filters: social sciences, psychology and education; languages: Spanish and English; text complete: yes; year of publication; 2018–2022 | 550 | 21 | 4 |
| | disabled student * AND (university OR higher education) | 226 | | 54 | | |
| | disabled student * AND (teacher attitude) AND (university OR higher education) | 104 | | 20 | | |
| | disabled student * AND teaching-learning process * AND innovation AND (university OR higher education) | 37 | | 9 | | |
| | disabled student * AND (difficult * OR obstacle *) AND (university OR higher education) | 38 | | 3 | | |
| | disabled student * AND (methodolog * OR program * OR intervention) AND (university OR higher education) | 83 | | 24 | | |
| | disabled student * AND (educational response) AND (university OR higher education) | 12 | | 2 | | |

* apocopated words.

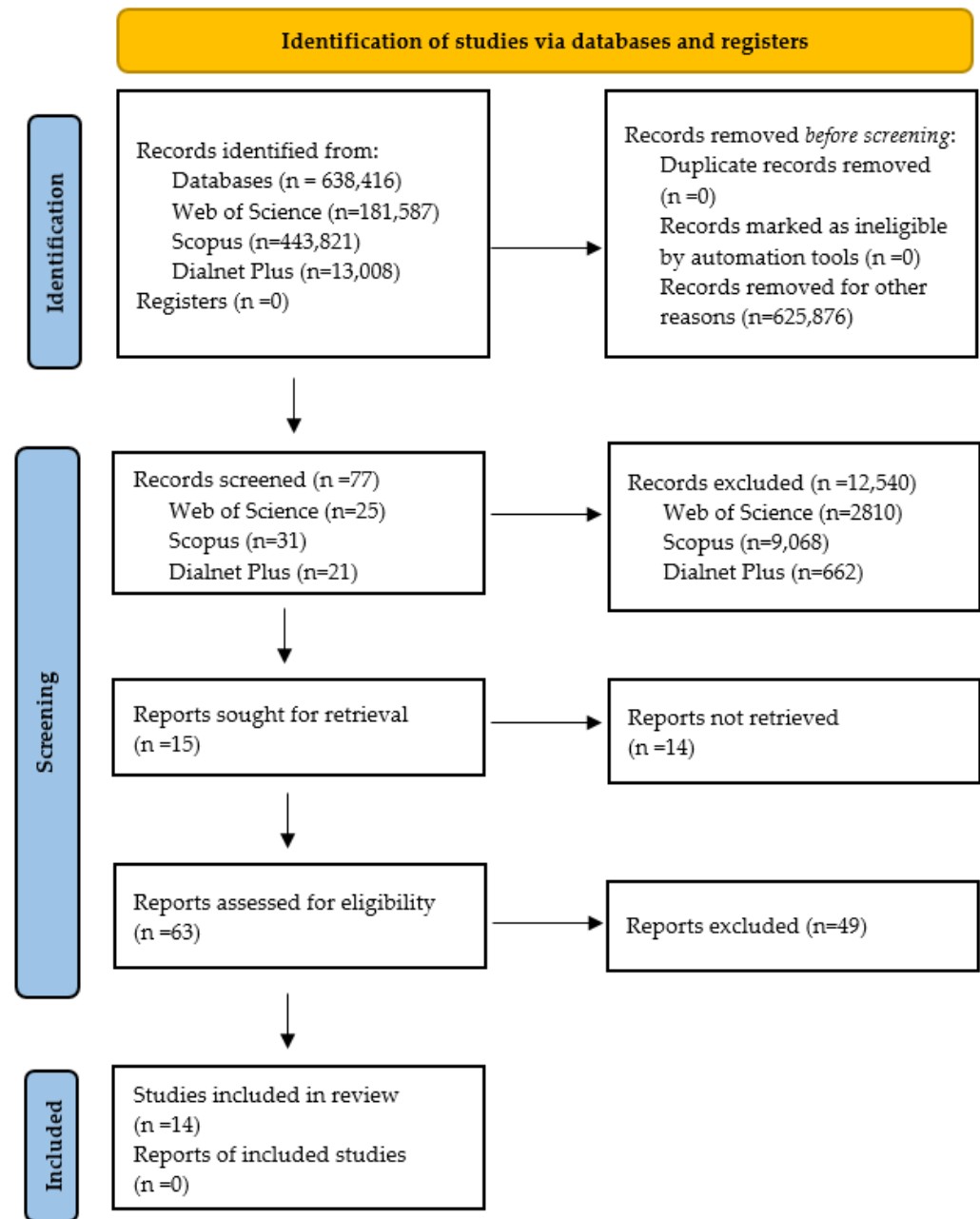

**Figure 2.** Selection Criteria Flowchart.

### 3. Results

This section presents the main characteristics of the articles that have been selected in this systematic review.

Initially, a total of 638,416 articles were found, which were the initial result of the search in the databases. After applying certain filters and language exclusion, a total of 77 papers were selected and read. Finally, after applying the rest of the exclusion criteria, a total of 14 investigations that met all the eligibility criteria were examined in depth (Table 2).

**Table 2.** General distribution of the selected works.

| Number | Author | Country | Research Objective | Sample | Sex | Age | Contact with People with Disabilities | Participants | Methodology | Teaching Attitude + = − | Strategies Used by Teachers |
|---|---|---|---|---|---|---|---|---|---|---|---|
| 1 | Díaz (2021) | Spain | Analyze the beliefs, necessary adjustments and difficulties of university professors | 42 | 40.5% men ($n = 17$) 59.5% women ($n = 25$) | Between 33 and 59 years, with the average being 41.2 years | Between 7 and 32 years of experience, the average being 15.8 years of teaching experience | University teachers from 6 public universities belonging to the faculty of education sciences | Qualitative. Bibliographic–narrative research with semi-structured individual interviews | X | Initial design according to the universal design for learning Attention to the emerging needs of students Adaptations in the subjects (content) Adaptations in the materials (texts, advance delivery of materials, text size) Architectural/furniture support Non-significant individualized adaptations, modifications in the teaching project Adjustments of times, activities, methodologies, type of evaluation |
| 2 | Polo Sánchez et al. (2021) | Spain | Analyze the beliefs and attitudes of university professors | 82 | 46.3% men ($n = 38$) 53.7% women ($n = 44$) | Between 31 and 40 years | Less than 5 years (28%); between 11 and 15 years (19.5%) | University teachers from public university belonging to the faculty of education sciences | Quantitative. Two questionaries: Scale of Attitudes towards People with Disabilities and Ideas and Attitudes on Skills, Training and Professional Development | X | |

**Table 2.** *Cont.*

| Number | Author | Country | Research Objective | Sample | Sex | Age | Contact with People with Disabilities | Participants | Methodology | Teaching Attitude | | | Strategies Used by Teachers |
|---|---|---|---|---|---|---|---|---|---|---|---|---|---|
| | | | | | | | | | | + | = | − | |
| 3 | Encuentra and Gregori (2021) | Spain | Analyze proposals for improvement in access to university | 421 | 49% men (*n* = 206) 51% women (*n* = 215) | 86.2% were adults 30–60 years old (30–35: 43.1%;45–59: 43.1%), 10.7% were below 29 years old and 3.1% were under 60 years old | Participants had different university access profiles (ranging from primary school education to PhDs) | Students with disabilities. Disabilities related to mobility (36.8%) and different diseases (25.9%) sensorial disabilities (14.9%) and mental illnesses (14.7%) | Quantitative. Survey based on the one used in an unpublished international comparison of access to e-learning for university students with disabilities. It comprised 24 closed-ended and four open-ended questions | | | | Time settings, activities, type of evaluation Technical and/or technological support Architectural/furniture support Teaching support Subject adaptations Adaptations in the subjects (content) Adaptations of the final tests |
| 4 | Lledó et al. (2020) | Spain | To analyze the application of inclusive methodologies in university | 313 | 64.2% men (*n* = 201) 35.8% women (*n* = 112) | | The years of teaching oscillate in a wide range of between one and forty years of teaching experience, with a mean of 14.81 and a standard deviation of 8.28 | University teachers; 39% belong to the field of social and legal sciences and 24.6% belong to the field of engineering and technology | Non-experimental quantitative approach characterized by the determination to quantify phenomena or opinions through numerical data without manipulating independent variables | | X | | Adaptations in the materials (texts, advance delivery of materials, text size) Methodology adjustments Use of inclusive methodologies Adjustments according to the universal design for learning |

**Table 2.** *Cont.*

| Number | Author | Country | Research Objective | Sample | Sex | Age | Contact with People with Disabilities | Participants | Methodology | Teaching Attitude + = − | Strategies Used by Teachers |
|---|---|---|---|---|---|---|---|---|---|---|---|
| 5 | Kendall (2018) | England | To explore the challenges for teachers to adapt to the needs of students with disabilities | 20 | 45% men ($n = 9$) 55% women ($n = 11$) | | They confirmed their experience with students with disabilities in both training and educational programs | University teachers from one faculty of a university | Qualitative. Interpretivist qualitative stance. Through semi-structured interviews | X | Intention to carry out inclusive practices and reasonable accommodations Attention to the needs of students Adaptations in the materials (color adjustments in the presentations, advance delivery of materials) Technological resources (audio recordings) Accommodations in assessments Writing support tutor |
| 6 | Collins et al. (2018) | Australia | To examine learning environments of students and the challenges faced by inclusive education | 40 | 50% men ($n = 20$) 50% women ($n = 20$) | More than 30 years old | Undergraduate and graduate students with disabilities | Students with disabilities and university teachers. * SWPD, $n = 11$; NDS, $n = 11$; AS, $n = 13$, two of whom were visually impaired; and DRC staff, $n = 5$ | Qualitative. Single case study with semi-structured interviews | X | Making reasonable adjustments Architectural/furniture support Technological resources (recordings) Support from other colleagues Time and evaluation settings |
| 7 | Bunbury (2018) | England | To analyze the inclusive curriculum and the duty to make reasonable adjustments | 5 | | | | University teachers from law school | Qualitative. An in-depth qualitative study based on interviews | X | Inclusive curriculum Methodology adjustments |

**Table 2.** *Cont.*

| Number | Author | Country | Research Objective | Sample | Sex | Age | Contact with People with Disabilities | Participants | Methodology | Teaching Attitude + | Teaching Attitude = | Teaching Attitude − | Strategies Used by Teachers |
|---|---|---|---|---|---|---|---|---|---|---|---|---|---|
| | | | | | | | | | | **Topic** | | | |
| 8 | Moriña et al. (2020) | Spain | To analyze the opinions of faculty members who carry out inclusive pedagogy | 119 | 58.33% men (*n* = 70) 41.66% women (*n* = 49) | | The majority had over 10 years (68.35%), only 6 had less than 5 years (6.25%) and 24 had between 5 and 10 years (25.4%). Sensory disabilities (visual or hearing impairment) were the most frequent (40.97%), followed by physical (23.68%), mental (18.79%) and poor-health conditions (10.52%), and learning difficulties (6.01%) | University teachers; 24 from art and humanities (20.16%), 14 from STEM (11.76%), 16 from health sciences (13.44%), 25 from social sciences and law (21.01%) and 40 from education science (33.61%) | Qualitative. A semi-structured interview | X | | | Attention to the needs of students Group settings, activities, methodologies, type of evaluation Feedback to students Subject adaptations |
| 9 | Langørgen et al. (2018) | Finland | To explore the perspectives on supporting disabled students in professional programs | 21 | 42.8% men (*n* = 9) 57.1% women (*n* = 12) | Age ranged from 35 to 65 | They state that they have experience in dealing with students with disabilities | University teachers (from health care, social work and teaching) and placement supervisors (from bachelor programs) | Qualitative. Based on focus group discussions (FGDs) with lecturers and placement supervisors | | X | | |
| 10 | Bartz (2020) | Germany | To re-examine the situation of disabled students in the university | 45 | 37.77% men (*n* = 17) 62.22% women (*n* = 28) | Aged 20 to 41 years with one or more disabilities | | Students with disabilities from 35 universities | Mixed methods. They were interviewed quantitatively (questionnaire) as well as qualitatively (narrative interviews) | | X | | |

Table 2. *Cont.*

| Number | Author | Country | Research Objective | Sample | Sex | Age | Contact with People with Disabilities | Participants | Methodology | Teaching Attitude | | | Strategies Used by Teachers |
|--------|--------|---------|--------------------|--------|-----|-----|----------------------------------------|--------------|-------------|:---:|:---:|:---:|------------------------------|
| | | | | | | | | | | + | = | − | |
| 11 | Moriña (2019) | Spain | To analyze motivation, emotion and the faculty–student relationships in the learning processes | 119 | 58.33% men (*n* = 70) 41.66% women (*n* = 49) | The majority was aged between 36 and 60, with seven (7.78%) being less than 35 years of age and four (4.42%) being over 60 | Most (68.35%) had over 10 years of experience, with only six (6.25%) having less than 5 and 24 (25.4%) having between 5 and 10 | University teachers from 10 Spanish universities; 24 (20.16%) taught arts and humanities, 14 (11.76%) taught STEM, 16 (13.44%) taught health sciences, 25 (21.01%) taught social and legal sciences and 40 (33.61%) taught education | Qualitative. Semi-structured interview | X | | | Attention to the needs of students Positive reinforcements Highly motivating strategies Plan teaching and learning processes Teacher training Classroom climate Active methodologies Adjustments in activities and resources Permanent feedback |
| 12 | Svendby (2020) | Norway | To look at attitudes and ideas about access to higher education | 5 | 40% men (*n* = 2) 60% women (*n* = 3) | | Each had at least eight years of teaching experience at the time of the interview with the exception of one person | University teachers. Their backgrounds cover the disciplines of social sciences, humanities, and technology | Qualitative. Interview | X | | | Attention to the needs of students Time adjustments, methodologies Adaptations in the materials (ppt presentations, reflection notes) Technical and/or technological support Communication with the teacher Instruction in inclusive practices to the rest of the students Collaborative works |

Table 2. *Cont.*

| Number | Author | Country | Research Objective | Sample | Sex | Age | Contact with People with Disabilities | Participants | Methodology | Topic | | | Strategies Used by Teachers |
|---|---|---|---|---|---|---|---|---|---|---|---|---|---|
| | | | | | | | | | | Teaching Attitude | | | |
| | | | | | | | | | | + | = | − | |
| 13 | Sulaj et al. (2021) | Albania | To analyze academic and access services | 148 | | | | Students with disabilities from 12 universities | Quantitative. Questionnaire in collaboration with Student Career Offices | | | | |
| 14 | Valle-Flórez et al. (2021) | Spain | To analyze the barriers that hinder educational inclusion | 201 | 44.4% men (*n* = 90) 55.7% women (*n* = 113) | The highest percentage in age range corresponds, in 41% of the cases, to professors over 50, more than half of the respondent sample (61%) is over 46; the youngest teaching staff (less than 30) group is the one with the slightest presence, specifically 6.7% | 58.1% have been working at the institution for more than 16 years, so we find a group with significant working experience; only 20% have experience of fewer than five years | University teachers from 2 public universities in the education faculties | Quantitative. It is a non-experimental, descriptive, and association design between variables using non-parametric techniques. | X | | | Adjustments of delivery times, activities, methodologies, resources and evaluation Modification of teaching resources Support from other colleagues Adjustments according to the universal design for learning Adaptations of the final tests Adaptations in the subjects (contents and objectives) |

* Notes: SWPD = students with physical disabilities, NDS = non-disabled students, DRC staff = disability resource center staff, AS = academic staff.

It should be noted that 64.28% of the selected works have been carried out using a qualitative methodology compared to 28.57% that have used the quantitative methodology and only 7.14% have used mixed methods, so the results allow deeply examining valuable information about the context, constructed meanings and personal experiences of the university professors.

To answer the first of the questions posed in this research (What is the attitude of university teachers about the need to provide an educational response to students with disabilities?), it is observed how the majority of teachers state that they have a positive attitude towards the presence of students with disabilities in the classroom (Díaz 2021; Polo Sánchez et al. 2021; Moriña 2019; Kendall 2018; Svendby 2020; Valle-Flórez et al. 2021; Collins et al. 2018; Bartz 2020).

The rise of this attitude means establishing a turning point in university teaching development, since by serving students with disabilities, a climate of support and understanding is generated that extends to all students, whether or not they have disabilities (Díaz 2021; Polo Sánchez et al. 2021; Valle-Flórez et al. 2021; Moriña et al. 2020). In this way, it is possible to establish the principle of equal opportunities and combat the generation of exclusionary contexts and promote the germination of spaces where differences between students are diluted (Kendall 2018; Svendby 2020; Collins et al. 2018). The presence of students with disabilities in university classrooms can be approached from several positive perspectives, on the one hand, adopting an empathic approach in which professional performance is humanized, and on the other hand, adopting an approach in which the disability of the student body is perceived as the opportunity to work with people who have different abilities (Svendby 2020).

This is reflected in the studies analyzed, where it is shown that 57.14% of the studies show positive attitudes of the teaching staff towards students with disabilities, except for the investigations by Langørgen et al. (2018) and Bartz (2020) that reveal as university professors have a negative attitude towards this type of student. They argued that the reasons are conditioned by individual cultural environments; by having to invest time, effort and responsibility in adapting the materials; and by the obligation to guarantee that students with disabilities have acquired the same skills as the rest of their class.

On the other hand, it is observed that in 14.28% of the works, there is no clear trend in the teaching attitude, since the teachers work with inclusive methodologies and adaptations in the subjects regardless of whether there is a student with a disability in the classroom (Bunbury 2018; Lledó et al. 2020).

In them, an attitude of distrust towards disability is perceived, since the teachers express the obligation of having to discern the veracity of the students' arguments about their disability and of having to analyze the needs of the student body, delaying their teaching action, which was influenced by the stigma towards the disability they manifested (Bunbury 2018). In addition, although their attitude became more positive, they insisted on the need for students with disabilities to acquire the same skills as the rest (Lledó et al. 2020).

The generation of a positive or negative attitude is sometimes influenced by the lack of knowledge that university professors have regarding disability in the classroom (Svendby 2020). Added to this situation is the lack of information and support that leads teachers to be unaware of this situation (Lledó et al. 2020). However, personal experiences in the classroom increase teachers' abilities to recognize and offer an educational response to the need for support (Polo Sánchez et al. 2021). However, the lack of positioning becomes a general awareness of diversity among all students (Bunbury 2018).

Regarding the second question (What innovative strategies do university professors use to develop teaching–learning processes with students with disabilities in university classrooms?), it is observed how the most used strategies to guarantee the teaching–learning processes have unequal results, and the implementation of different inclusive strategies such as the modification or adaptation of methodologies (Díaz 2021; Bunbury 2018; Lledó et al. 2020; Moriña 2019; Valle-Flórez et al. 2021; Moriña et al. 2020), contents (Díaz 2021; Moriña 2019; Svendby 2020; Valle-Flórez et al. 2021; Encuentra and Gregori 2021) or other

aspects related to their subject turns out to have a heterogeneous application (Díaz 2021; Bunbury 2018; Lledó et al. 2020; Kendall 2018; Encuentra and Gregori 2021).

The analyzed works reveal the strategies developed in university classrooms in favor of carrying out inclusive practices that enable the participation of all students, including people with disabilities as a valuable element within the learning context. Some of the strategies present in the studies analyzed are characterized by being actions that sought to motivate the entire student body, whether they are disabled or not. Among them, the following stand out:

1.  Reasonable accommodations and classroom climate, where the organization of the classroom and access to the facilities are essential to generate a warm, close environment and belonging to the group (Moriña 2019). However, on many occasions, this access to the classroom has been conditioned by factors external to the teachers (Kendall 2018; Svendby 2020). Reasonable adjustments are mainly based on the need to attend to the diversity of all the students present in the classrooms and in this way promote and guarantee the principle of equal opportunities and generate non-discriminatory contexts that do not imply differences between the students (Díaz 2021; Collins et al. 2018; Moriña et al. 2020).

2.  Adaptation of teaching materials is another of the most recurrent strategies used by university professors since it facilitates access to content and objectives for all students (Lledó et al. 2020; Valle-Flórez et al. 2021; Encuentra and Gregori 2021). However, this strategy continues to generate controversy in the face of the attitude of the teachers and that of the rest of the classmates, denoting negative positions towards the generation of content presentations adapted to sensory disabilities (Kendall 2018) or the refusal to make recordings of the plenary sessions to facilitate their understanding (Bunbury 2018; Lledó et al. 2020; Langørgen et al. 2018; Bartz 2020).

3.  Active and inclusive methodologies are promulgated as another of the strategies that have been carried out by university teachers in order to deal with traditional education represented by the master class (Encuentra and Gregori 2021) and enhance the learning carried out by students by increasing understanding, motivation and participation of the same (Valle-Flórez et al. 2021). This strategy requires specific training by university teachers to include them in the classroom (Moriña 2019). However, it is a very positive option since it encourages the participation and commitment of students and teachers to develop an affable attitude towards disability, leaving aside the traditional methodologies so present in university environments (Díaz 2021; Banks 2019; Kendall 2018; Collins et al. 2018).

4.  Adaptations of resources and activities: Didactic resources and materials are a valuable tool to generate inclusive spaces; the use of PowerPoint presentations, reflection notes (Bunbury 2018), providing notes and materials before classes (Lledó et al. 2020; Valle-Flórez et al. 2021; Encuentra and Gregori 2021), starting by making a brief reminder of the previous class (Lledó et al. 2020) and encouraging students to investigate and broaden their knowledge of the topic discussed (Díaz 2021; Collins et al. 2018) are some examples which provide support for all students and therefore reduce the need to establish reasonable adjustments in the development of the teaching activity (Díaz 2021; Bunbury 2018). These educational strategies encourage a greater degree of participation in students and allow the generation of a wide range of resources (Kendall 2018; Collins et al. 2018; Encuentra and Gregori 2021) that facilitate learning not only for students with disabilities but also for the rest of the university students. Group dynamics are presented as an inclusive strategy where instead of working on disability individually, they work from a group approach (Lledó et al. 2020; Valle-Flórez et al. 2021). The solution to the problems is configured in a community way, and the creation of an inclusive climate is encouraged where the leaders of each group receive training in disability matters to promote inclusive activities and develop the potential of all students (Svendby 2020).

5. Individual tutorials and continuous feedback: Individual tutoring favors the establishment of human connections in relation to disability and inclusion, not requiring specific training for its development, and manages to favor a positive attitude towards disability and seek solutions to small didactic problems (Lledó et al. 2020; Valle-Flórez et al. 2021). The adoption of this type of action shows a closer and positive attitude towards disability that facilitates the generation of continuous and clear progress where those aspects of the teaching and learning processes that need specific support will be reinforced, in addition to generating safe spaces. where students can share their concerns (Kendall 2018; Svendby 2020). The importance of immediate feedback favors the generation of more significant knowledge (Moriña 2019).

6. Adaptations in the evaluation processes: This is one of the most widely adopted strategies in university classrooms since it allows for longer times in final tests and does not require specific training to be able to carry it out (Díaz 2021; Lledó et al. 2020; Moriña 2019; Kendall 2018; Moriña et al. 2020; Encuentra and Gregori 2021). Among the strategies adopted, the extension of the time in the final tests (Valle-Flórez et al. 2021), the use of technical or technological supports (Encuentra and Gregori 2021) and the use of alternative evaluation methods (Díaz 2021; Moriña et al. 2020) stand out.

The research methodology carried out requires the establishment of a restriction of years in order to be able to know and analyze the most current research on the teaching attitude, the strategies used in the classroom and the main difficulties in offering an educational response to university students with disabilities. Figure 3 shows how the evolution of the number of articles that deal with these topics has been according to the established period of analysis.

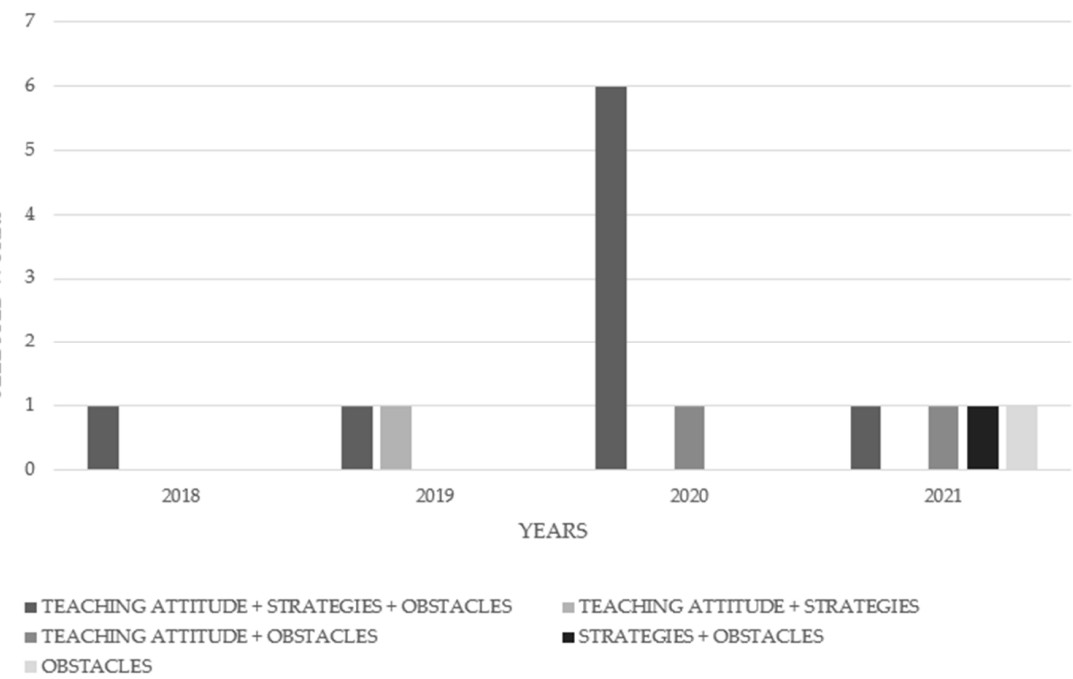

**Figure 3.** Number of works selected in relation to the year and theme treated.

It is observed that in the year 2020, there are a large number of studies that analyze the educational response towards students with disabilities, examining the three variables in this study: teaching attitude, classroom strategies and obstacles. This demonstrates the great interest in providing a quality educational response to students with disabilities at universities. This finding suggests that the educational response to university students with disabilities continues to be an interesting research area for the scientific community. As for the rest of the variables, a homogeneous distribution is observed.

Finally, answering the third question (What are the main obstacles when attending to students with disabilities in universities?), Table 3 shows the main difficulties that have been detected when offering an educational response to university students with disabilities, from the perspective of university professors and students with disabilities.

**Table 3.** Main obstacles in the educational response to university students with disabilities.

| Participants | Difficulties Detected |
|---|---|
| University teachers | Application of inclusive methodologies<br>Assumption of other roles<br>Architectural barriers, non-adapted furniture<br>Invisibility of disability<br>Curriculum Design<br>Lack of time<br>Training of university teachers in the field of attention to disability<br>Traditional methodologies<br>Few resources |
| Students with disabilities | Teaching attitude<br>Architectural barriers, non-adapted furniture<br>Lack of teacher training<br>Lack of materials in the classroom<br>Lack of specific transport services<br>Lack of time to do homework<br>Problem of access to didactic/academic materials<br>Computer system accessibility issues<br>Administrative tasks<br>Universities without attention to people with disabilities |

It is observed how architectural barriers or non-adapted furniture (Sulaj et al. 2021; Valle-Flórez et al. 2021; Moriña et al. 2020; Bartz 2020) and lack of teacher training (Díaz 2021; Bunbury 2018; Kendall 2018; Valle-Flórez et al. 2021; Collins et al. 2018; Langørgen et al. 2018)are common difficulties that both groups reveal. In addition, various obstacles related to the application of inclusive methodologies are highlighted (Lledó et al. 2020), those related to materials, resources and services (Sulaj et al. 2021; Polo Sánchez et al. 2021; Svendby 2020; Collins et al. 2018; Bartz 2020; Encuentra and Gregori 2021) in higher education are some of the main difficulties in promoting the full inclusion of students with disabilities in universities.

Teacher training in disability matters continues to be a turning point since, on the one hand, teachers state that this training should be mandatory and regular (Kendall 2018), while others advocate establishing training courses on basic knowledge that provide security when working with this type of student (Collins et al. 2018). The lack of teacher training has negative effects that directly affect planning and collaboration to alleviate current needs (Valle-Flórez et al. 2021).

The wide range of disabilities present in university classrooms generates feelings of frustration in teachers as they feel overwhelmed and unable to offer a quality educational response (Collins et al. 2018; Moriña et al. 2020). Support services for students with disabilities continue to generate conflicting opinions; on the one hand, they are perceived as a help, and on the other, personal experiences show how the instructions received by this service are general or inappropriate with the reality of their students, so they are forced to carry out individualized tutoring to be able to specify the necessary support (Kendall 2018) or to resort to unreliable sources (Google) to be able to solve their needs (De la Rosa Ruiz et al. 2019). This idea is present in a large number of the studies analyzed (Díaz 2021; Bunbury 2018; Polo Sánchez et al. 2021; Collins et al. 2018; Moriña et al. 2020; Langørgen et al. 2018; Encuentra and Gregori 2021).

The lack of pedagogical resources is a common difficulty that causes teachers to feel disconcerted by the need to provide an educational response to this type of student

(Bunbury 2018; Lledó et al. 2020). However, in this situation, it is possible to act using the baggage that a teacher has created with the development of educational experiences, or a feeling of inaction can be generated due to the lack of knowledge to be able to solve this situation (Díaz 2021). This generates great difficulty when it comes to serving students with disabilities (Moriña et al. 2020).

Academic barriers or obstacles are also present within the virtual institutions, since in the analyzed articles it is shown how the problems of accessibility to the system and the didactic materials and the lack of time to carry out the tasks are very present (Encuentra and Gregori 2021).

Another barrier present in the studies analyzed is the lack of time for both students and teachers to face the required tasks and respond to the expressed needs (Langørgen et al. 2018; Encuentra and Gregori 2021), to which is added the need to assume, on occasions, the role of therapist to ensure that students achieve their pedagogical development (Svendby 2020). Students with disabilities require a wide variety of resources for the normal development of teaching, from basic aids (Díaz 2021; Moriña 2019) to specialized teaching materials (Valle-Flórez et al. 2021; Collins et al. 2018). Having academic materials in advance is also shown with our strategy to not be exempt from detractors since the advancement of these resources implies that they must be created in advance, generating in the teaching staff a situation of stress and pressure in the face of this fact (Kendall 2018; Svendby 2020; Encuentra and Gregori 2021).

Another of the most common difficulties within the promulgation of any methodology is related to the evaluation processes that, despite the good predisposition of teachers to make reasonable adjustments, show how this educational practice is becoming difficult. to maintain (Kendall 2018).

Finally, the invisibility on the part of the student with disabilities, for fear of negative actions (Langørgen et al. 2018; Bartz 2020) or not feeling judged or pitied (Bunbury 2018), is why they choose the distance university (Encuentra and Gregori 2021), leaving the face-to-face university. Hiding the disability either in the virtual or face-to-face environment entails greater difficulties when it comes to offering necessary adjustments in the evaluation processes because there is no margin to act (Kendall 2018). The decision to communicate the disability or not is free, although the literature shows how it is conditioned by the associated stigma (Riddell and Weedon 2014; Verdinelli and Kutner 2016).

## 4. Discussion

In this article, a systematic review of the most recent studies and research on the educational response offered to students with disabilities in higher education institutions has been carried out. A comprehensive set of 14 articles has been analyzed.

In this context, the following research questions have been raised: (1) What is the attitude of university teachers about the need to provide an educational response to students with disabilities? (2) What innovative strategies do university professors use to develop teaching–learning processes with students with disabilities in university classrooms? (3) What are the main obstacles when dealing with students with disabilities in universities? These three questions are linked to the more general objective of this study, which is to present critical information about the educational response offered to students with disabilities in higher education institutions.

This general objective is broken down into the following specific objectives: (1) analyze current attitudes of university teaching staff after years of inclusive processes; (2) identify innovative strategies developed by university professors to carry out the teaching–learning processes with students with disabilities in university classrooms; (3) describe what remain the main obstacles or difficulties in dealing with students with disabilities in universities.

### 4.1. Attitude of Teachers towards Students with Disabilities in University Classrooms

In relation to the first objective, a large number of investigations were evidenced that yielded positive results in terms of teaching attitude (Díaz 2021; Polo Sánchez et al.

2021; Moriña 2019; Kendall 2018; Svendby 2020; Valle-Flórez et al. 2021; Collins et al. 2018; Moriña et al. 2020). These results are linked to the statements offered in various studies in which it is shown that the success of inclusive education is conditioned by the willingness and open, proactive, flexible and receptive attitude of teachers (Moriña and Carballo 2020; Biewer et al. 2015).

Higher education institutions have the duty to offer access facilities to students with disabilities, and it is the responsibility of the entire educational community to promote the necessary actions to make this possible (Polo Sánchez et al. 2021). This idea of social responsibility is related to the results of other studies (Norris et al. 2019; Sutton-Long et al. 2016) where it is stated that society must establish structural organizations in which the characteristics and needs of all people are taken into account.

The analyzed works that yielded negative results with respect to the teaching attitude (Langørgen et al. 2018; Bartz 2020) present curious data. In both works, the results shown from the teaching perspective (Langørgen et al. 2018) and from the student body (Bartz 2020) argue that dealing with teachers sometimes becomes complex by denying access to different academic activities and even manifesting exclusionary attitudes towards the refusal to adapt evaluation processes or relate them to racist attitudes.

These results are in line with other studies carried out in which it is revealed that the negative attitudes of university teachers generate feelings of disinterest, stress or overload in the teachers themselves (Sánchez 2016), and this favors the generation of little motivation to care for these students and decreased curiosity in participating in training programs or information on how to care for students with disabilities in the classroom (Rodrigues 2015). However, this situation is further aggravated if possible, when finding studies that point to the existence of teachers who show their negative attitudes towards the inclusion of students with disabilities in the initial educational stages, since they emphasize the large number of necessary adjustments that must be made., the excessive time to provide individualized attention, the lack of support services or the quality of the work carried out by these students (de Boer et al. 2012; Gallegos Navas 2017). All these negative attitudes have a direct impact on teaching processes and are externalized towards elements such as not prioritizing curricular adaptations to meet the needs of students (Ortiz Colón et al. 2018) or insufficient pedagogical competence regarding values such as tolerance or empathy around this type of students (Ke et al. 2017).

On the other hand, the two investigations analyzed in which the teaching attitude was neither positive nor negative (Bunbury 2018; Lledó et al. 2020) show significant differences with respect to the professional category; young teachers develop more positive attitudes towards disability. These results are related to previous studies in which it was stated that young teachers had had greater contact with innovative and inclusive methodologies in their teacher training and therefore had greater sensitivity, awareness and predisposition to develop positive attitudes (Hellmich and Görel 2014; Urton et al. 2014). These results are in line with other works analyzed in this systematic review (Díaz 2021; Polo Sánchez et al. 2021; Moriña 2019; Svendby 2020; Valle-Flórez et al. 2021; Collins et al. 2018; Moriña et al. 2020).

Finally, the study developed by Bunbury (2018) shows that teachers with negative attitudes were aware that they have a stigma associated with disability that conditioned their educational practices. This teaching perception causes a complexity when it comes to establishing a balance between differential and similar treatment among university students, which together with the non-inclusive conditions of the institutions considerably affects the teaching attitude, as has been observed in the study developed by Edwards et al. (2022).

### 4.2. Innovative Strategies to Promote Teaching–Learning Processes within University Classrooms

In relation to the second objective, teaching styles in higher education institutions are varied and multidisciplinary. Within the university teaching–learning processes, the most important piece continues to be the teacher (Bagnato 2017). Adapting methodologies, processes and content continues to be a challenge for university professors (Moriña 2019; Valle-Flórez et al. 2021). Reasonable adjustments, as their name indicates, must be in

accordance with the present disability (Collins et al. 2018). A good teaching predisposition sometimes generates negative feelings in professionals such as feeling under pressure (Díaz 2021), being overwhelmed or fearful that the adjustments will not work and being accused of discrimination (Svendby 2020). These adjustments affect the broader aspects of the teaching–learning processes, which implies that training in this area must be greater (Kendall 2018). The results analyzed agree with those shown in other studies (Lightfoot et al. 2018; Banks 2019; Aguilar et al. 2019; Sandoval et al. 2020).

The development of inclusive strategies is largely conditioned by the attention to diversity that is intended to be given in each subject. In previous educational stages, the development of inclusive teaching–learning processes is characterized by the fact that teachers know in advance the presence of students with disabilities in the classroom (Urton et al. 2014). In research carried out in the university field, the identification of disability has conflicting results. Fear of stigma (Edwards et al. 2022), teaching attitude (Langørgen et al. 2018; Bartz 2020) or peer perceptions (Riddell and Weedon 2014; Akin and Huang 2019) causes less than 50% of students to reveal their disability (Riddell and Weedon 2014; Bartz 2020; Newman and Madaus 2014; Fossey et al. 2017). However, other studies analyzed (Sulaj et al. 2021) show the immediacy with which students communicate their situation in order to receive the necessary adaptations. This situation is a very significant conditioning factor, as manifested in other works analyzed (Svendby 2020; Encuentra and Gregori 2021), because when applying inclusive practices in the university classroom, the lack of knowledge on the part of teachers about the presence of students with disabilities is an influential factor in the present needs not being met (Banks 2019; Valle-Flórez et al. 2021).

The implementation of an inclusive curriculum involves developing flexible objectives that are in line with active methodological planning and participation in which the role of the students is greater than that of the teacher (Moriña 2019; Moriña et al. 2020). As has been seen in the studies analyzed, in the university setting, its implementation requires a proactive teaching attitude and a modification in the curricular framework of all subjects in order to meet the needs of students with disabilities (Bunbury 2018). As a consequence, the need to make reasonable adjustments would be reduced, and this would provoke the incursion of inclusive educational practices where different methodological styles could be alternated (Collins et al. 2018). This perspective is related to other studies in which the commitment to an inclusive curriculum following the guidelines of the universal design for learning will allow providing the necessary answers to the needs raised (Hayward et al. 2020).

In relation to the universal design for learning, similar results were obtained in various investigations in which it is stated that teachers tend to create flexible learning scenarios within the classrooms depending on the methodological strategies that are adopted, with the creation of cooperative groups being one of the most used strategies (Strnadová et al. 2015; Burgstahler 2015).

The application of active and inclusive methodologies causes teachers to have greater flexibility with respect to the time of carrying out the teaching–learning processes where adapting the teaching materials is a simple strategy that ensures that the teaching processes are adapted to different learning rhythms (Polo Sánchez et al. 2021; Lledó et al. 2020). However, the lack of training in active and inclusive methodologies (Bunbury 2018; Moriña 2019; Kendall 2018; Valle-Flórez et al. 2021; Collins et al. 2018) and the application of different strategies such as the use of the cooperative group (Kendall 2018) or individual tutoring (Svendby 2020) represent a weakness in higher education institutions when presenting diverse learning styles as demonstrated in other developed studies (Aguilar et al. 2019; Blinova et al. 2022). However, it is observed how the university educational paradigm has overcome the generation of segregating contexts and has already changed the mentality of teachers towards a more positive perspective (Díaz 2021; Polo Sánchez et al. 2021; Moriña 2019; Kendall 2018; Svendby 2020; Valle-Flórez et al. 2021; Collins et al. 2018; Moriña et al. 2020).

The adaptations carried out in the evaluation processes are other strategies adopted in the university environment, both face-to-face and virtual (Encuentra and Gregori 2021). The importance of achieving the objectives proposed in the curriculum and the development

of valid and reliable competencies does not disagree with the execution of alternative evaluation methods (Langørgen et al. 2018). In contrast, the analyzed studies (Bunbury 2018; Langørgen et al. 2018) stand out; they reveal the refusal of teachers to carry out alternative evaluations of students with disabilities based on the design of the study plan, the importance of learning results or developing doubly exclusive attitudes when relating disability with racist comments.

*4.3. Current Obstacles to Serving University Students with Disabilities*

Finally, with regard to the third objective, according to the results of the studies analyzed in this systematic review, in higher education institutions, incessant improvements are being produced with respect to accessibility and mobility of students with disabilities, but they are still not sufficient to ensure a full inclusion (Díaz 2021; Bunbury 2018; Sulaj et al. 2021; Polo Sánchez et al. 2021; Lledó et al. 2020; Kendall 2018; Svendby 2020; Valle-Flórez et al. 2021; Collins et al. 2018; Moriña et al. 2020; Langørgen et al. 2018; Bartz 2020). The exhaustive analysis developed by Sulaj et al. (2021) shows that both university infrastructure and services do not meet the needs of students with disabilities. There are still architectural barriers, and although adaptations and remodeling have been carried out, they have not been enough to create accessible routes. This result differs from the original proposal of universal design, in which before its transfer to the educational field, it was thought of as a measure to develop an accessible architecture (Meyer et al. 2014). Within the architectural barriers there is a special section for inappropriate furniture, which generates feelings of frustration by depending on elements that escape teacher control (Kendall 2018; Svendby 2020; Moriña et al. 2020). A possible solution to this paradigm lies in the creation of a team of professionals made up of therapists, politicians, engineers and architects who coordinate efforts to build university buildings that are more accessible to people with disabilities (Sarsak 2018).

Regarding the accessibility of the routes, the results shown by Sulaj et al. (2021) agree with those presented by Cepeda et al. (2018) in which it is shown that public transport is one of the essential services for students with disabilities to access universities.

Other difficulties, present in most of the works analyzed, and which have been expressed by teachers and students, are related to academic barriers (Díaz 2021; Bunbury 2018; Polo Sánchez et al. 2021; Lledó et al. 2020; Kendall 2018; Svendby 2020; Valle-Flórez et al. 2021; Collins et al. 2018; Langørgen et al. 2018; Bartz 2020; Encuentra and Gregori 2021). The scarce teacher training in the field of disability is shown as one of the most present difficulties in the studies analyzed in this review (Díaz 2021; Bunbury 2018; Sulaj et al. 2021; Polo Sánchez et al. 2021; Kendall 2018; Valle-Flórez et al. 2021; Collins et al. 2018; Langørgen et al. 2018). The need to train teachers in the field of disability means advancing towards an education that is increasingly less segregating and permissive in order to expand the methodologies used in the classroom towards flexible approaches oriented under the guidelines of inclusive pedagogies (Bunbury 2018; Sulaj et al. 2021). These results coincide with those stated by Williams et al. (2019) where instructing academic professionals on disability issues and innovative educational practices positively affects the accessibility and inclusion of students with disabilities.

A clear example that would serve university teachers in their purpose of continuing to increase their knowledge on disability is related to the teacher training programs that are configured as small voluntary and free training courses on various topics of interest to university teachers (Gunersel and Etienne 2014) that, distributed throughout the entire academic year, offer the possibility of increasing knowledge (Simpson 2002) about disability, inclusion, universal learning design, methodologies, etc. (Cunningham 2013; Moriña and Carballo 2018). However, for these to be effective, they must be correctly evaluated to affirm that they are really effective and help university professors understand the importance of offering an inclusive educational response.

The wide range of disabilities and associated needs is frustrating for university professors who say they feel overwhelmed in providing an educational response to these students.

Support services for students with disabilities continue to generate conflicting opinions as they are perceived as a resource that does not meet the needs of teachers (Díaz 2021; Bunbury 2018; Polo Sánchez et al. 2021; Collins et al. 2018; Moriña et al. 2020; Langørgen et al. 2018; Encuentra and Gregori 2021). These results are related to other developed studies (Edwards et al. 2022; Perera et al. 2022; Rillotta et al. 2022) showing that the lack of collaboration generates a serious difficulty in establishing an inclusive education in the university environment.

However, despite the negative perceptions observed in the studies analyzed, the support offices for students with disabilities within the universities are configured as a key element when it comes to guiding teachers and students about the academic–administrative organization, providing information and disability education (Aller and Villa 2011). In this way, these services must be consolidated as a space aimed at the entire university community that promotes awareness with this group and becomes a forum for inclusion (Moliner García et al. 2019).

The virtual barriers or obstacles (Encuentra and Gregori 2021) manifested in the results obtained are in line with other research where it is stated that virtual environments are not as accessible as one might think and require adaptations and substantial changes to be used by all students autonomously (Kutscher and Tuckwiller 2019; Lucas Barcia et al. 2022; Jacob et al. 2022; Melián and Meneses 2022).

The material and personal resources turn out to be of significant importance for carrying out the teaching–learning processes. Their scarcity generates feelings of frustration in university teachers (Bunbury 2018; Langørgen et al. 2018). According to the studies analyzed in this review, many of these materials are not always adapted to the needs of students with disabilities or generate discomfort in the rest of their classmates (Sulaj et al. 2021; Bartz 2020). These results complement those exposed by Kendall (2018) who states that using teaching materials such as PowerPoint presentations or recordings of the sessions generates situations of inhibition in teachers and students regarding participation in class.

Another barrier present in the studies analyzed is the lack of time (Langørgen et al. 2018), which is likely to generate negative attitudes towards these students (Bartz 2020). The excessive bureaucracy of higher education institutions generates frustration both for teachers, seeing that the time needed to make reasonable adjustments is reduced (Bunbury 2018), and for students with disabilities who negatively perceive the excessive procedures and administrative fees for them to be able to pursue their studies (Bartz 2020; Encuentra and Gregori 2021).

Another limiting barrier to inclusive education at a university is related to the difficulty in using active and inclusive methodologies that allow students to fully participate in the teaching and learning processes. It has been observed that traditional education extolled by lectures (Blinova et al. 2022) continues to be present in the university context (Lledó et al. 2020). As a solution to the proposed paradigm, other investigations bet on using inclusive tools and strategies that generate non-exclusive spaces and gradually dilute the present barriers (Mena et al. 2018). In this sense, the commitment to facilitate accessibility and increase teaching–learning processes based on universal design would reduce the dependence that inclusion has on the attitudes of teachers (Mayán 2017).

All these academic barriers make up a network of difficulties that has a common axis in the need to propose more inclusive study plans where the application of active methodologies, evaluations, adaptations of materials, academic practices, contents and objectives does not pose a constant challenge for teachers who are frustrated and overwhelmed by the need to adapt, with little room for action, their educational practices to the needs of students with disabilities (Bunbury 2018; Bartz 2020).

Limitations and Suggestions

The development of this research was configured from the beginning with the firm purpose of presenting a general overview with relevant information on what are the attitudes of university teachers, such as the methodological strategies that are used, and

what are the main obstacles to solve in the university inclusion of students with disabilities. This may be a limitation in not focusing the study on one type of disability, but at the same time, this research establishes the general premises for a new line of research in which the focus is on one type of disability.

In general terms, the studies analyzed are characterized by the fact that the samples have been collected due to the interest shown by the subjects in participating in the study, leading to the conjecture that very few university professionals agree to participate in these investigations regarding their attitude towards the educational practices carried out in the classroom. This greatly hinders the purpose of research in general, which is to detect problems in society and investigate solving them. In this way, a bias can be perceived in that all the professionals who considered themselves to have a positive attitude participated, and a few, under guaranteed anonymity, offered to chat openly about topics that generated interest.

Another limitation is the lack of literature that deals with the educational response in general in university institutions, because although there are many publications that talk about architectural barriers or lack of resources, those that deal with the attitude of university teachers are very restricted. It is just as important to resolve those obstacles that prevent physical access to universities as it is to solve the academic problems that generate negative attitudes.

For this study, the PRISMA guidelines were followed to be able to complete each of the sections and report the data obtained; in this way, it should be considered that the studies and investigations that have been analyzed present diverse methodologies, both quantitative and qualitative. This fact can be perceived as a strength and also as a limitation; however, the methodological variety makes it possible to carry out an analysis with a greater breadth of guidelines that are being carried out in the different universities to serve students with disabilities. The results shown in this systematic review have been obtained by performing a general comparison of the different findings obtained in the selected investigations, without establishing a meta-analysis. Therefore, the result of this systematic review can serve as a basis for future meta-analyses and empirical investigations, always keeping in mind the aforementioned limitation.

It is vitally important to continue studying the characteristics of the different educational systems in relation to the educational response provided to students with disabilities to establish the possible differences that each one of them presents.

In the present study, the great variety of opinions, perceptions and experiences around how the teaching–learning processes take place in university classrooms has been revealed; in this way, knowing what the strong points are and what the weak points are within university environment can serve as an aid to improve the educational systems in each country.

## 5. Conclusions

The group of people with disabilities has traditionally been one of the vulnerable groups that has seen the most limited access to and permanence in certain social, employment and educational services. It is understood, in this case, that people with disabilities are included within this term when they encounter special difficulties in fully exercising their rights and freedoms or when encountering obstacles in their personal, social, labor and educational development. This has been reflected in many countries where the necessary supports, reasonable adjustments and curricular adaptations that allow equal opportunities and thus eradicate inequalities associated with disability are still not offered. This has been latent in the various indicators of inclusion of university students, where imbalances are observed, for example, in access to higher education. However, not everything is negative for this group, since it has also been observed how many countries have been pioneers when it comes to making progress in this area since they have been able to eradicate some of the main barriers to learning, and consequently, active participation of this group in various educational and social actions has been promoted. Nevertheless, the commitment to inclusive education in higher education institutions continues to be an objective to be achieved by all educational systems. The systematic review of the literature carried out in this work has revealed an

objective reality: universities are still not prepared to provide a quality educational response to all their students regardless of their characteristics and needs.

The teaching attitude continues to be a backbone to promote the promulgation of an inclusive education within the university environment, and although education should not be conceived as the mere transmission of knowledge in a unidirectional way between teacher and student, the traditional methodologies, for which master classes are the greatest representatives, continue to plague the educational spaces of higher education.

However, there are many teachers who, faced with an increasingly diverse reality, do not promulgate the approach of traditional education and opt for the application of various educational strategies that promote the development of more inclusive teaching and learning processes.

For this to be possible, it is necessary to solve some of the difficulties and obstacles that are still present both in the university context and in the academic field, namely that the traditional architectural barriers are joined by the lack of training of university professors in the matter of attention to disability; the refusal to adapt contents, materials or objectives to the characteristics of the students; and the difficulties professors face in accessing the materials and carrying out the activities.

Offices or support services for disabled students, together with university teacher training programs, have been configured as two elements that, characterized by their heterogeneity, currently, on the one hand, act as mediators between teachers and students and make it possible to create centers of socialization, inclusion, orientation and student and teacher support (support services) and, on the other hand, act as training tools for teachers that provide the necessary resources to carry out necessary adaptations and adjustments that allow progress and permanence of students in these institutions (university teacher training programs).

This review reinforces the idea that it is necessary to continue advancing in terms of inclusion in the university environment, promoting spaces for communication and establishing relationships between teachers and students, the incessant need to increase training on disability for university teachers and the elimination and eradication of architectural barriers that hinder face-to-face access to the university.

A line of future research may be related to those positive aspects that are developed in the various inclusive educational practices in university classrooms and that require more research and dissemination.

University professors who are committed to inclusive education have experienced the beneficial effects of this type of action, and that the rest of the community should have the opportunity to analyze it.

Higher education institutions have the opportunity to seriously consider the challenges and opportunities posed by providing an educational response to students with disabilities in order to improve the social, employment and educational inclusion of this group.

**Author Contributions:** Conceptualization, M.D.P.-E. and J.J.C.-M.; methodology, M.D.P.-E.; formal analysis, J.J.C.-M.; investigation, L.O.J. and J.J.C.-M.; writing—original draft preparation, M.D.P.-E.; writing—review and editing,. L.O.J. and J.J.C.-M.; visualization, M.D.P.-E.; supervision, J.J.C.-M.; project administration, J.J.C.-M.; funding acquisition, all authors. All authors have read and agreed to the published version of the manuscript.

**Funding:** This research was supported by the Spanish Ministry of Science, Innovation and Universities (Ministerio de Ciencia, Innovación y Universidades) Within the framework of the FPU program (PhD fellowship) granted to the first author M.D. P-E. with reference FPU18/05887.

**Institutional Review Board Statement:** Not applicable.

**Informed Consent Statement:** Not applicable.

**Data Availability Statement:** Not applicable.

**Conflicts of Interest:** The authors declare no conflict of interest.

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
