# Peer review of "Systematic Review on New Challenges of University Education Today: Innovation in the Educational Response and Teaching Perspective on Students with Disabilities"

_socsci, doi:10.3390/socsci12040245_

Round 1

Reviewer 1 Report

I am really very thankful for receiving this article for reviewing.

The article is very interesting, conclusive and well structured .

The aim was to present the systematic review carried out following the PRISMA method.

The main objective was to characterise the university teachers attitudes and the support dedicated to the students with disabilities at the tetrially education level. The main interests were given to teaching attitude, difficulties and strategies used in higher education institutions.

Particular scientific questions were as following:

What is the attitude of university professors about the need to provide an educational response to students with disabilities?

What strategies do university professors use to develop teaching-learning processes with students with disabilities in university classrooms?

Introduction – The general background in the field of inclusive education at university level and the legal statements at the Unated Nations and European context were given as well as the examples of the countries (Spain, UK, Argentina, Albania) were universal accessibility or non-discrimination of people with disabilities at university level were legally protected. The selection of the mentioned countries is confusing – no logical criterion. In many other countries the process of inclusive education implementation at university level has been started, eg Poland. So I would suggest to add references (and others):

ZieliÅ„ska, M. (2022). The Policy of Supporting Students with Disabilities At Higher Education Institutions in Poland — The Example of the University of WrocÅ‚aw University and WrocÅ‚aw University of Technology. Polish Political Science Review, 10, 1, 97-113 DOI: https://doi.org/10.2478/ppsr-2022-0006

 Method – the description of the PRISMA method used is clear.

Particularly noteworthy is the number of initially analyzed articles selected because of the assumed key concepts. A total of 638,416 articles were found  in the databases (WoS, SCOPUS, Dialnet plus).  Finally, after applying the rest of the exclusion criteria, a 207 total of 14 investigations that met all the eligibility criteria were examined in depth. The key concept "disabled student" as the including criterion is too wide to give particulare information on the adjustment needs.

Results – the results were presented based on the 14 articles analysed, in a clear manner adequate to the research questions posed. The main description in the context of the replies to the first and second research questions mixed are given in the text including percentage ratiost (lines 223-257). It would be more communicative to present answer to the each scientific question separately.

However there is the lack of tabular presentation of the analysis of articles on university teachers’ attitudes. The presentation of the attitudes - the general description of the method of the research described in the particular articles should be add (research instruments used, the characteristics of the study group). General cathegory (university teachers) is insufficient. The results of the research considered in the PRISMA analysis can be interpreted differently depending on the individual characteristics of the group studied (age, gender, seniority, contact with people with disabilities). Very important is the precise description of the instrument had been used by particulare cited authors, because instruments may have a different structure, refer to different theoretical concepts - one, two or three-factors concepts of attitudes) then a nominal comparison of attitudes indicators would be illegitimate.

The main aspects of university teachers inclusive strategies were presented (table 2). In my opinion the description should also extended with additional data about the instrument - interview?, and group characteristic).

Disccusion  - the discussion was led smoothly with reference to the materials confirming the importance of university teachers attitudes and the their positive level. There is the lack of literature on negative attitudes of university teachers (but the low percentage of teachers interested in participating in the survey may provide a rationale for assuming that these teachers might have negative attitudes - it was pointed out in the limitations part) however there are some reference of teachers of special schools who are usually against fool school inclusion. These teachers are the most familiar with the special needs as well as the reality of individual adjustment so they might be also found as the context of results interpretation.

Conclusions and limits – there is the lack of special characteristic of particular disability: sensory, mental health, ASD, ID, LD. In my opinion the project unjustifiably limits the analysis to the general characteristics of the persons with disabilities, without taking into account the specific needs and requirements . It is impossible to abstract from these variables, as we come to the misleading conclusion that almost everything in university teaching practice depends on the attitudes of teachers, their training and ability to apply strategies. Without limiting everyone's right to education, however, we should consider whether the intellectual and behavioral qualities of the individual person provide a rationale for undertaking the university education.

Author Response

Dear reviewer, thank you very much for your contributions that will undoubtedly help to improve our manuscript. The changes addressed appear below and are highlighted in the original document in green.

Introduction – The general background in the field of inclusive education at university level and the legal statements at the Unated Nations and European context were given as well as the examples of the countries (Spain, UK, Argentina, Albania) were universal accessibility or non-discrimination of people with disabilities at university level were legally protected. The selection of the mentioned countries is confusing – no logical criterion. In many other countries the process of inclusive education implementation at university level has been started, eg Poland. So I would suggest to add references (and others):

ZieliÅ„ska, M. (2022). The Policy of Supporting Students with Disabilities At Higher Education Institutions in Poland — The Example of the University of WrocÅ‚aw University and WrocÅ‚aw University of Technology. Polish Political Science Review, 10, 1, 97-113 DOI: https://doi.org/10.2478/ppsr-2022-0006

The selection of the countries initially mentioned, (Spain, United Kingdom, Argentina and Albania) was produced by presenting extensive experience in the promulgation of accessibility policies and non-discrimination of people with disabilities. However, and taking into account your recommendation, this list of countries has been updated incorporating Poland, Chile and Australia as example countries of the promulgation of this type of non-discriminatory actions. You can see the changes in the lines (33-51)

 Method – the description of the PRISMA method used is clear.

Particularly noteworthy is the number of initially analyzed articles selected because of the assumed key concepts. A total of 638,416 articles were found  in the databases (WoS, SCOPUS, Dialnet plus).  Finally, after applying the rest of the exclusion criteria, a 207 total of 14 investigations that met all the eligibility criteria were examined in depth. The key concept "disabled student" as the including criterion is too wide to give particulare information on the adjustment needs.

In the initial construction of this systematic review of the literature, various topics were raised on which to focus this study, but from its genesis we were very aware that we wanted to offer a general idea about the attitudes, strategies and obstacles that arise within the university context to students with disabilities. Due to this, this descriptor "student with disabilities" was chosen as a key term when carrying out the searches and as an inclusion criterion.

However, given his observation and the antecedents reflected in this research, we believe that it would be very interesting to focus on a type of disability (sensory, TEA, ID, LD) to carry out future research and be able to contrast these results.

Results – the results were presented based on the 14 articles analysed, in a clear manner adequate to the research questions posed. The main description in the context of the replies to the first and second research questions mixed are given in the text including percentage ratiost (lines 223-257). It would be more communicative to present answer to the each scientific question separately.

Dear reviewer, the presentation of the two scientific questions has been addressed separately, in this way, the results obtained are shown more clearly and without confusion. You can see the changes made to the (232-236) lines for the first question and to the (290-293) lines for the second question.

However there is the lack of tabular presentation of the analysis of articles on university teachers’ attitudes. The presentation of the attitudes - the general description of the method of the research described in the particular articles should be add (research instruments used, the characteristics of the study group). General cathegory (university teachers) is insufficient. The results of the research considered in the PRISMA analysis can be interpreted differently depending on the individual characteristics of the group studied (age, gender, seniority, contact with people with disabilities). Very important is the precise description of the instrument had been used by particulare cited authors, because instruments may have a different structure, refer to different theoretical concepts - one, two or three-factors concepts of attitudes) then a nominal comparison of attitudes indicators would be illegitimate.

The main aspects of university teachers inclusive strategies were presented (table 2). In my opinion the description should also extended with additional data about the instrument - interview?, and group characteristic).

Dear reviewer, Table 2 has been modified to add the requested information that makes it possible to quickly and clearly understand the main characteristics of the selected articles. Especially in relation to the teaching attitude and inclusive strategies, since the information regarding the research method has been expanded, the main characteristics of the group, such as sex, specification of the subjects (university teachers), since not in all the selected studies specified seniority, age or contact with people with disabilities.

In this way, and thanks to your recommendation, we believe that with the modifications made to the information in Table 2, they make it possible to complement the information described in the results section.

Disccusion  - the discussion was led smoothly with reference to the materials confirming the importance of university teachers attitudes and the their positive level. There is the lack of literature on negative attitudes of university teachers (but the low percentage of teachers interested in participating in the survey may provide a rationale for assuming that these teachers might have negative attitudes - it was pointed out in the limitations part) however there are some reference of teachers of special schools who are usually against fool school inclusion. These teachers are the most familiar with the special needs as well as the reality of individual adjustment so they might be also found as the context of results interpretation.

In response to his recommendation, studies and research have been expanded that corroborate that university teachers have certain negative attitudes towards students with disabilities, although as mentioned in the limitations section, this has been a great handicap to overcome. You can observe the modifications in the lines (475-488)

Conclusions and limits – there is the lack of special characteristic of particular disability: sensory, mental health, ASD, ID, LD. In my opinion the project unjustifiably limits the analysis to the general characteristics of the persons with disabilities, without taking into account the specific needs and requirements . It is impossible to abstract from these variables, as we come to the misleading conclusion that almost everything in university teaching practice depends on the attitudes of teachers, their training and ability to apply strategies. Without limiting everyone's right to education, however, we should consider whether the intellectual and behavioral qualities of the individual person provide a rationale for undertaking the university education.

Dear reviewer, as previously mentioned, in the initial configuration of this research, the development of the study was proposed from a general perspective in order to provide an overview of what teacher attitudes are like, what are the methodologies that are applied in the classroom and which are the main obstacles in the university context. It is because of this, that neither in the limitations nor in the conclusions is special mention made of some of the variables associated with disability. However, and thanks to the appreciation in your comment, we have considered it necessary to include this clarification in the limitations section to avoid confusion for readers. You can observe this modification in the lines (652-658)

Reviewer 2 Report

The paper provides a systematic review of the research involving higher education response to students with disabilities. The discussion is well set out and clearly explains its intent and objectives. The two main matters raised from the review were:

1. Be careful of making generalisations. For example, line 43 claims there has not been an increase in disabled people enrolling in higher education. That claim can be disputed in different countries e.g., Australia.

2. Justification is not given regarding the choice of English and Spanish parameters in the literature search.

The paper should provides readers with several important issues to consider in their engagement with disability in higher education.

Author Response

Dear reviewer, thank you very much for your contributions that will undoubtedly help to improve our manuscript. The changes addressed appear below and are highlighted in the original document in blue.

  1. Be careful of making generalisations. For example, line 43 claims there has not been an increase in disabled people enrolling in higher education. That claim can be disputed in different countries e.g., Australia.

The wording has been modified to avoid generalizations of this type, since, as he very well points out, an increase in the enrollment of this type of student body has been observed. You can observe the modifications made in the lines (56-61)

  1. Justification is not given regarding the choice of English and Spanish parameters in the literature search.

The justification of the parameters in English and Spanish has been added in section 2.1. Phase 1; search strategies. You can observe this modification in the lines (170-172)

The paper should provides readers with several important issues to consider in their engagement with disability in higher education.

The formulation of this systematic review in its genesis has the firm purpose of offering readers relevant information on three crucial aspects when talking about inclusive education that takes place in the university environment. These aspects are; the attitudes of university teachers; the innovative strategies that are carried out in university classrooms and; the main obstacles that are perceived regarding the development of this inclusive education.

We believe that these issues are very important when it comes to advancing the development of an inclusive university education that advocates for continuing to engage with disability in higher education.

Reviewer 3 Report

It is a good article in general terms and I recommended the publication but with minor changes. The objectives of the research are really interesting as they are giving visibility to students with disabilities in our universities and they still have serious problems to be accepted and receive teaching in the best way possible, They should foster legal regulations and university organizations if so to help them. There are offices in different universities to help them when they have problems with the teachers and not only with the building, According to my experience and my own university, I know that some teachers do not help them to study because it is supposed that they are not going to work in the future and this is a clear phobia of teachers and not a problem of training. So you have to explain alternatives to offer the right to the pupils to study all they want. Maybe some Training Programs for Teachers should be obligatory to attend correctly this collective of pupils. They should explain better the concept of Vulnerable Groups. It seems that they are guilty because of the way they are. Teachers are afraid to teach them or think that they are vulnerable, oppression means that the group with the power to make decisions can classify them in this way. The article has a high quality.

Author Response

Dear reviewer, thank you very much for your contributions that will undoubtedly help to improve our manuscript. The changes addressed appear below and are highlighted in the original document in yellow.

There are offices in different universities to help them when they have problems with the teachers and not only with the building, According to my experience and my own university, I know that some teachers do not help them to study because it is supposed that they are not going to work in the future and this is a clear phobia of teachers and not a problem of training. So you have to explain alternatives to offer the right to the pupils to study all they want. Maybe some Training Programs for Teachers should be obligatory to attend correctly this collective of pupils.

Based on your recommendation, we have proceeded to explain in greater detail the aid offices for university students with disabilities (lines 608-613), as well as the teacher training programs (593-600). In addition, these considerations have been mentioned in the conclusions section in the lines (729-736)

They should explain better the concept of Vulnerable Groups. It seems that they are guilty because of the way they are. Teachers are afraid to teach them or think that they are vulnerable, oppression means that the group with the power to make decisions can classify them in this way. The article has a high quality.

Dear reviewer, thank you very much for your recommendation, we have proceeded to clarify the concept of "vulnerable group" since it is not our intention to blame this group for their way of being. You can see this clarification in the lines (697-708)

Round 2

Reviewer 1 Report

Thank you very much for improving your article.

Well done. Congratulations!

Joanna Kossewska